# Unreliable Monte Carlo Dropout Uncertainty Estimation

Aslak Djupskås[1], Signe Riemer-Sørensen[*2], and Alexander Johannes Stasik[1,2]

[1]Department of Data Science, Norwegian University of Life Sciences, Ås, Norway
[2]SINTEF AS, Department of Mathematics and Cybernetics, Oslo, Norway

## Abstract

Reliable uncertainty estimation is crucial for machine learning models, especially in safety-critical domains. While exact Bayesian inference offers a principled approach, it is often computationally infeasible for deep neural networks. Monte Carlo dropout (MCD) was proposed as an efficient approximation to Bayesian inference in deep learning by applying neuron dropout at inference time [1]. Hence, the method generates multiple sub-models yielding a distribution of predictions to estimate uncertainty. We empirically investigate its ability to capture true uncertainty and compare to Gaussian Processes (GP) and Bayesian Neural Networks (BNN). We find that MCD struggles to accurately reflect the underlying true uncertainty, particularly failing to capture increased uncertainty in extrapolation and interpolation regions as observed in Bayesian models. The findings suggest that uncertainty estimates from MCD, as implemented and evaluated in these experiments, is not as reliable as those from traditional Bayesian approaches for capturing epistemic and aleatoric uncertainty.

## 1   Introduction

In numerous practical applications, particularly those where decisions have significant consequences, machine learning models need to provide not only accurate point predictions, but also reliable estimates of their uncertainty [2]. Understanding when a model is uncertain about its prediction is vital for safe and robust deployment.

Bayesian machine learning offers a principled approach to quantifying uncertainty by modelling probability distributions over model parameters or functions [3]. Under perfect conditions, this framework allows for the capture of epistemic uncertainty (due to lack of knowledge, potentially reducible with more data) and aleatoric uncertainty (inherent noise in the data) [4]. However, performing exact Bayesian inference in complex models like deep neural networks is computationally intensive and often impractical [5] even with modern improvements such as full batch Hamiltonian Monte Carlo [6].

To address the computational challenges of

Bayesian deep learning, Monte Carlo dropout (MCD) was proposed as a more efficient approximation [1, 7]. The method involves applying dropout not only during training, but also at inference (prediction) time. By performing multiple forward passes in parallel with different dropout masks, one effectively samples from an ensemble of thinned networks, and the variance of these predictions is used as an estimate of model uncertainty. It is claimed that this method is mathematically equivalent to an approximation of a probabilistic deep Gaussian process. The approach has been challenged from various angles, including the interpretation as Bayesian [8] and unexpected behaviour and strong sensitivity on dropout parameter choice [9] as well as empirical bad comparison to true Bayesian posteriors [10, 11]. The main goal of this work is to empirically investigate the behaviour of the MCD method on simple regression problems. We perform a series of controlled experiments and compare to two Bayesian benchmark models: Gaussian Processes (GP) and Bayesian Neural Networks (BNN) which converge to identical posteriors in the large-layer regime [12]. The focus is on the ability to capture uncertainty during interpolation (small gaps in data within the training range) and extrapolation (outside the training data range).

## 2   Related Work

Bayesian methods offer a framework for modelling uncertainty [3]. Instead of learning fixed parameters, Bayesian models infer probability distributions over parameters. Gaussian Processes (GP) are non-parametric Bayesian models that define a distribution directly over functions [13, 14]. For any finite set of input points, the corresponding function values have a joint Gaussian distribution, defined by a mean function and a covariance function (kernel). GPs naturally provide predictive distributions, capturing uncertainty based on the training data and kernel choice. The process involves computing a posterior distribution given training data and optimizing kernel hyperparameters, often by maximizing the marginal likelihood.

Bayesian Neural Networks (BNN) extend standard neural networks by placing probability distributions over their weights and biases, rather than learning single point values [15, 16]. Computing the

[*]signe.riemer-sorensen@sintef.no

Proceedings of the 7th Northern Lights Deep Learning Conference (NLDL), PMLR 307, 2026.

exact posterior distribution over weights in BNNs is generally intractable, necessitating approximation methods. Common approaches include **variational inference (VI)** or **sampling methods** where the posterior is approximated through samples that converge to the target distributions. Common sample methods include Markov Chain Monte Carlo (MCMC) and Hamiltonian Monte Carlo (HMC).

The posterior predictive distribution of BNNs can be expressed as

$$
\begin{aligned}
p(\tilde{y} \mid \tilde{x}, z_n) &= \int_{\Theta} p(\tilde{y} \mid \tilde{x}, \theta) p(\theta \mid z_n) d\theta \\
&= \mathbb{E}_{\theta \sim p(\theta \mid z_n)}[p(\tilde{y} \mid \tilde{x}, \theta)] \quad (1)
\end{aligned}
$$

where $\tilde{x}$ is the new input and $\tilde{y}$ is the corresponding output, $z_n$ is the training data with $n$ samples and $\theta$ are the learned parameters. The latter equality corresponds to a marginalising over the learnable parameters which for an over-parametrised network smears out the epistemic uncertainty encoded in $p(\theta \mid z_n)$. Hence, single BNNs may not reliably reproduce the epistemic uncertainty [17–19].

**Monte Carlo Dropout (MCD)** was suggested as a computationally efficient alternative for obtaining uncertainty estimates in deep learning [1]. The core idea is to keep dropout layers active during inference and perform multiple stochastic forward passes. Each pass uses a different dropout mask, effectively sampling from a distribution of sub-networks. The mean of these samples provides the prediction, and their variance quantifies the model's uncertainty. The original paper suggests that this is equivalent to a Bayesian approximation for deep Gaussian processes [14]. However, this relation can be questioned as MCD can be interpreted as a non-Bayesian modal approximation as it introduces a sparsity inducing prior [20].

For machine learning models targeting decision purposes, the epistemic uncertainty is less relevant than the total uncertainty with respect to the decision objective [21]. However, epistemic uncertainty is still relevant to understand model performance and data needs. Here we perform an empirical comparison of predicted uncertainties from MCD, BNN, and GPs on a simple regression problem and demonstrate that even in data-rich scenarios, the predicted uncertainties are very different.

## 3    Methodology

The investigation of MCD was conducted using controlled regression tasks with synthetic data described in Section 3.1 and 3.2. The data was generated with a known noise distribution allowing for direct comparison between the model predictions and the true generating function together with the known noise characteristics.

In 1D we compare MCD with GP and BNN on datasets with varying amounts of training data (15, 50, 150 points) and noise levels ($N(0, 0.05)$ and $N(0, 0.2)$). A gap in training data $[-0.5, 0.5]$ was included to test interpolation and extrapolation. For each method, the mean and 95% confidence interval of the predictions are compared to the true underlying function. The experiment also included analyzing the impact of different random seeds on MCD's predictions. In 2D we compare MCD with GP for a noise level of ($N(0, 0.05)$).

### 3.1    1D Toy problem data

Synthetic data was generated from a known one-dimensional function:

$$
f(x) = \sin(4x) + r_0 + r_1 x + (r_2 + {}^1\!/{}_2)x^2, \quad (2)
$$

where $r_0, r_1, r_2$ are sampled from $\mathcal{N}(0, 1)$ and the input data was sampled from the range -1.3 and 1.3. The training data excludes the interval $[-0.5, 0.5]$ to specifically evaluate interpolation performance. We performed noise-free experiments to ensure that the model was expressive enough that we could ignore any aleatoric uncertainty. The noise was sampled from a normal distribution $\mathcal{N}(0, \sigma_{obs}^2)$, with constant $\sigma_{obs}$ across the range or scaled with $\sin(x)/x$. Lastly, the data set was standardised.

### 3.2    2D Toy problem data

For the 2D experiments we define a scalar field with three components i) a smooth base field, ii) high frequency radial oscillations in the center, iii) a 'ring of Gaussian beads'. See Figure 4 upper left. The training data were sampled from outside the circle such that the model never has seen the radial oscillations. The noise was sampled from a normal distribution with constant $\sigma_{obs} = 0.05$ across the field.

### 3.3    Implementations

The implementations are available on github[1].

**The Monte Carlo Dropout (MCD)** was implemented in PyTorch and Keras/Jax, using a custom MCDLayer class. The network architecture consisted of an input layer, two hidden layers with ReLU activation, and an output layer. The dropout was applied during the forward pass at inference time to obtain multiple predictions. The mean and variance of these predictions provided the estimate of the predictive distribution. In the 1D experiments, we used hidden layers of 32 neurons each, $L_2$ regularization of $10^{-4}$, learning rate of $10^{-3}$ and a dropout inference rate of 0.15 on the hidden layers. With small number of input features (three),

---

[1] https://github.com/aslakdjupskas/MCDO

the dropout removal of units risks disrupting the input structure, and adding noise too early may cause instability. Dropout in the output layer introduces randomness into the final prediction, making uncertainty estimates harder to interpret. In contrast, dropout in hidden layers effectively explores different sub-models of the decoded input, before encoding it back to the predicted output. In the 2D experiments, we use two hidden layers of 124 neurons each and otherwise the same parameters as in the 1D experiments.

**The Gaussian Process (GP)** was implemented with GPJax [22] using a zero-mean prior and Radial Basis Function (RBF) kernel, with a Gaussian likelihood assuming independent noise. Hyperparameters were optimised by minimizing the negative marginal log-likelihood using the Adam optimiser.

**The Bayesian Neural Network (BNN)** was only applied to the 1D case. It was implemented in NumPyro [23, 24] with a two-hidden-layer architecture with tanh activation. Weights were assigned normal priors, and observation noise had a Gamma prior. The No-U-Turn Sampler (NUTS) was used to sample weights [25] during inference. Predictions were made by averaging outputs from posterior weight samples, typically without adding observation noise during inference to focus on model uncertainty.

**Hyper parameter optimization** was performed using Optuna and then kept fixed for all experiments.

# 4 Results

## 4.1 Comparing methods 1D

Figure 1 compares the means and 95% confidence intervals of the 1D predictions provided by GP, BNN, and MCD. The results are shown for increasing numbers of training data points (15, 50, 150). For all three frameworks, the increase in data points leads to an improved approximation (lower MSE) of the true function, particularly in the range of the training data, but also in the interpolation range.

The GP and BNN models show a characteristic increase in the confidence interval 95% in regions without training data, i.e. in the interpolation gap $[-0.5, 0.5]$ and in extrapolation regions outside the training data range $[-1, -0.5]$ and $[0.5, 1]$. This reflects increasing uncertainty when the model is asked to predict values far from observed data. In contrast, the MCD dropout model exhibits a more constant variance across the entire input range without pronounced increase in interpolation or extrapolation intervals.

When increasing the shot noise to $N(0, 0.2)$ (Figure 2), the variances increase significantly in the data region for both GP and MCD, while the BNN

variance remains similar to the $N(0, 0.05)$ scenario. In the interpolation region, the variance increases for all frameworks, but for the MCD, the variance is still similar across all regions, without increasing in the data gap.

These results suggest that, in this comparative setting, MCD's inherent mechanism does not automatically yield an uncertainty estimate that captures the varying levels of confidence (epistemic uncertainty) in regions further from training data as effectively as the benchmark Bayesian models.

It is known that random initialisations of neural networks may lead to different outcomes, especially with weak regularization [26]. Figure 1 and 2 only visualise one such realisation. Performing each experiment four times with different seeds showed that while initialisation lead to different mean solutions, the behaviour of the variance is consistent across individual initialisations (Figure A.1). In addition, the model was trained using 100 different seeds resulting in a mean MSE of 0.384 and with a MSE standard deviation of 0.156.

Table 1 quantify the difference in posterior across different seeds.

| Seed | MSE | Mean STD Error |
|------|--------|----------------|
| 5 | 0.1436 | 0.0113 |
| 6 | 0.3230 | 0.0172 |
| 7 | 0.3325 | 0.0106 |
| 8 | 0.1564 | 0.0157 |

**Table 1.** MSE and Mean STD errors for different seeds.

Figure 3 shows that combining the posterior predictions from different random seeds for MCD can result in an uncertainty profile that more closely resembles that of the GP and BNN models, with increased uncertainty in extrapolation and interpolation areas. This is because the variations across different seeds are more pronounced in data-scarce regions, but this requires training an ensemble of MCD models thereby significantly increasing the computational load.

## 4.2 Comparing methods 2D

Figure 4 shows the results for the 2D task. The number of training samples were selected to achieve similar fit quality in the training data leading to 500 training samples for the GP and 10000 for the neural network with MCD, and a shot noise of $\sigma = 0.05$. Notice how the GP has narrow variance in the regions where the data are sampled and higher variance in the hole, while the MCD has constant variance across the range, but high residuals in the hole, which is consistent with the behaviour observed in 1D.

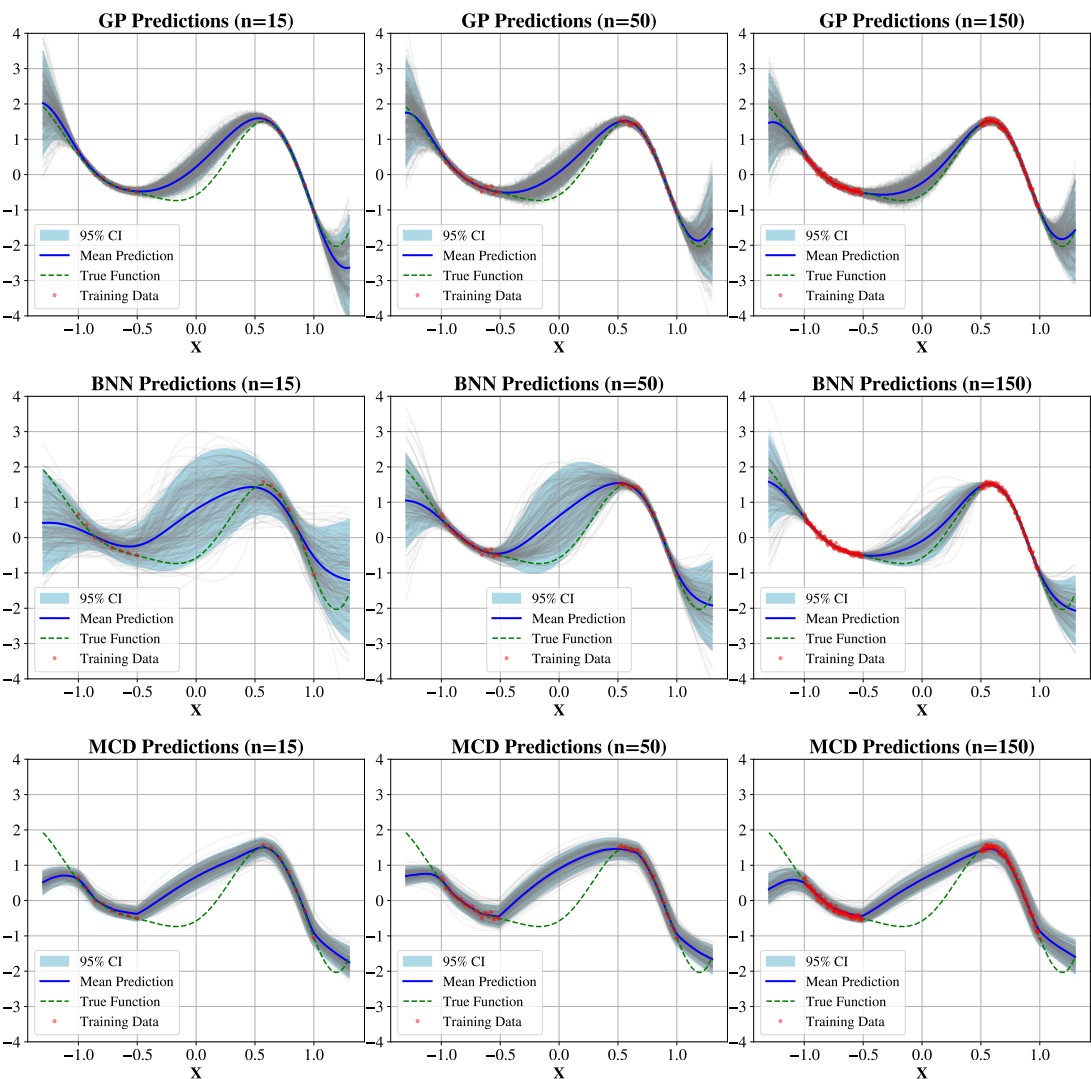

**Figure 1.** 1D results: Comparing the Gaussian Process (GP, top row), Bayesian Neural Network (BNN, middle row) and Monte Carlo Dropout (MDC, bottom row) when fitted to 15, 50 and 150 data points (left, middle, right, respectively) with a shot noise of $\sigma = 0.05$. Notice how the GP and BNN has narrow variance in the regions where the data are sampled and higher variance in the interpolation and extrapolation regions, while the MCD has constant variance across the range.

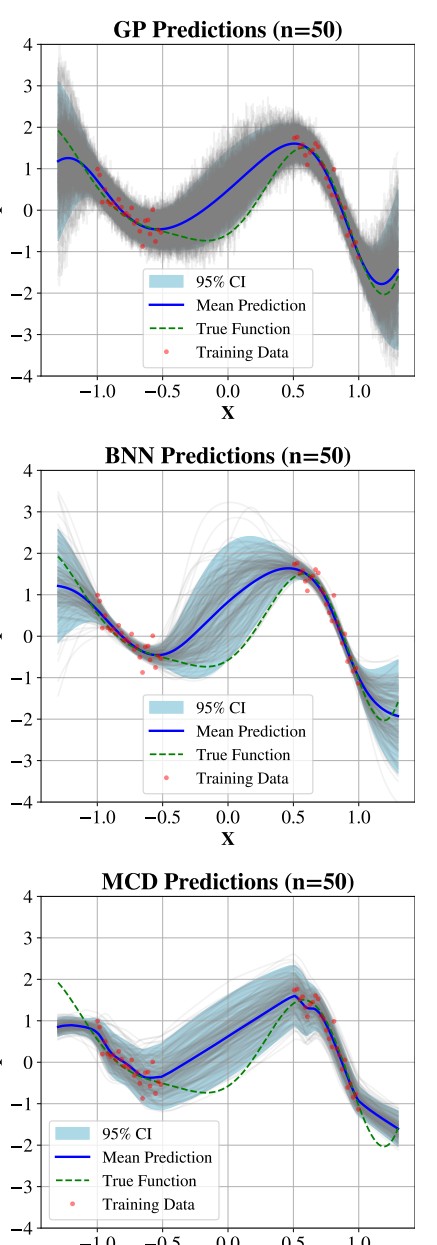

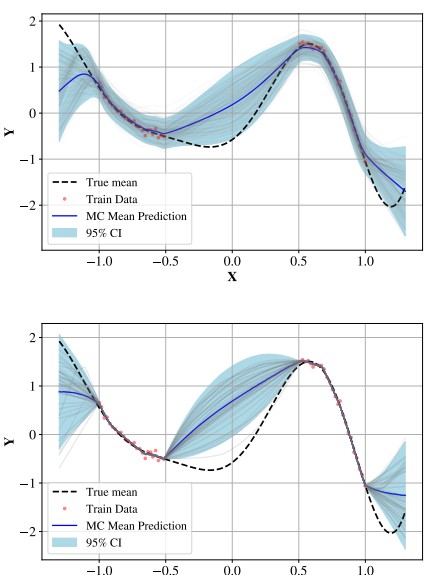

**Figure 3.** Random initialisations of the network also creates variation. In the example, a MCD model is fitted to 50 data points with a shot noise of $\sigma = 0.05$. Combining predictions trained with different initializations, leads to a profile that more closely resembles that of the GP and BNN models (upper panel, compared to second column of Figure 1). In the top panel, the dropout is applied in training, while in the bottom panel dropout is not applied, and gives a more similar shape as the Bayesian methods.

**Figure 2.** Comparing the Gaussian Process (GP, top row), Bayesian Neural Network (BNN, second row) and Monte Carlo Dropout (MDC, third row), when fitted to 50 data points with a shot noise of $\sigma = 0.2$.

## 5  Discussion

Our experimental results, provide insight into the behaviour of MCD as an uncertainty estimation method and allow us to address the research question regarding its ability to approximate true uncertainty.

Comparing MCD with the benchmark Bayesian models (GP and BNN) reveals a conceptual difference in how uncertainty is captured. GP and BNN models naturally show increasing uncertainty as predictions move away from training data points (extrapolation and interpolation regions), reflecting

a decrease in epistemic certainty due to lack of information. This behaviour is a desirable property for reliable uncertainty estimation.

In contrast, the MCD tends to produce a more uniform uncertainty across the input space. This suggests that MCD's uncertainty estimate, primarily derived from the variance of predictions across different dropout masks, may not be effectively capturing epistemic uncertainty in the same way that GP or BNN models do. Predicting with high confidence in areas far from training data or regions with large noise, where the model has limited evidence, is a significant drawback for safety-critical applications.

The dependency of MCD's posterior on the random seed further questions its reliability as an objective uncertainty estimator. An ideal uncertainty model should not have its uncertainty profile heavily influenced by the specific initialization or randomness during training or inference. While combining predictions from different seeds can create a more Bayesian-like uncertainty profile, this suggests that a single MCD model might not be sufficient, potentially requiring ensemble modelling, which adds computational cost.

Based on the claim that MCD is mathematically equivalent to an approximation to a deep Gaussian process [1] we expected similar uncertainty char-

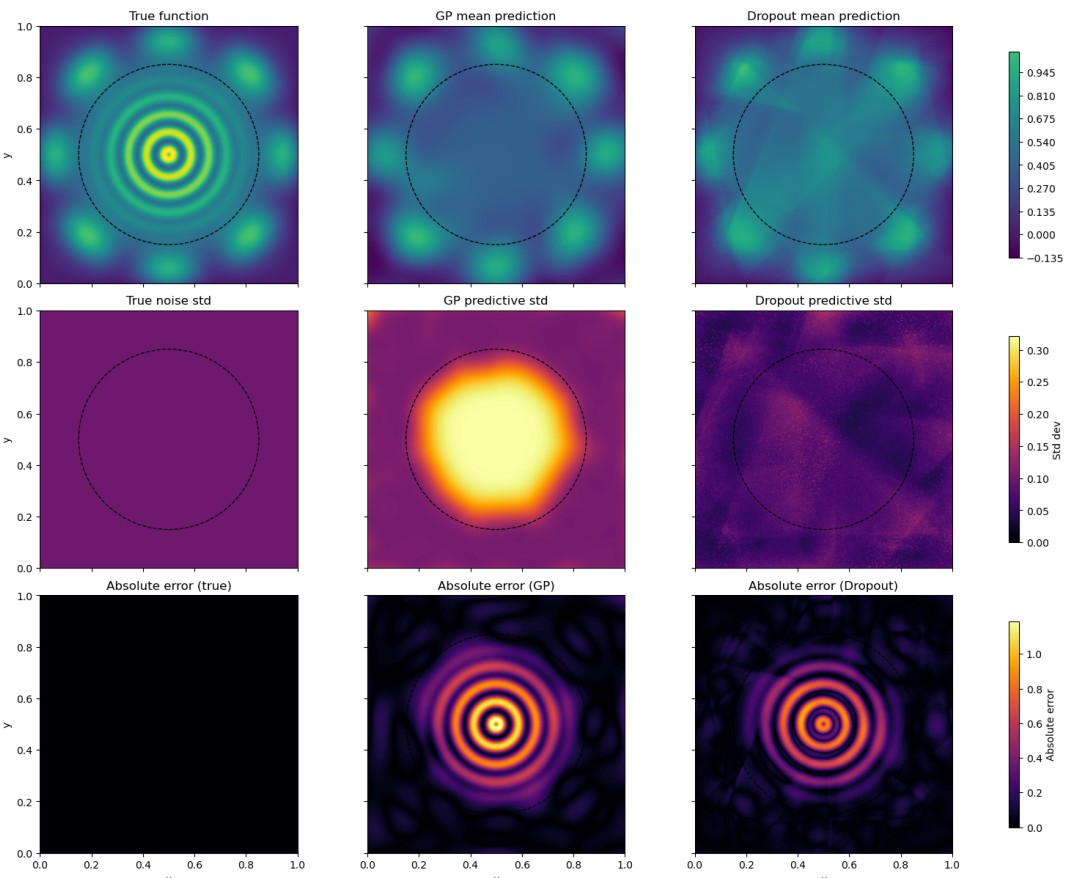

**Figure 4.** Comparing the predictions (top row), variance (middle row) and residuals (bottom row) for 2D generated data (left column), Gaussian Process (GP, middle column), and Monte Carlo Dropout (MDC, right column). The number of training samples were selected to achieve similar fit quality in the training data leading to 500 training samples for the GP and 10000 for the neural network with MCD, and a shot noise of $\sigma = 0.05$. Notice how the GP has narrow variance in the regions where the data are sampled and higher variance in the hole, while the MCD has constant variance across the range, but high residuals in the hole.

acteristics, particularly increased uncertainty away from data. Our results, showing a generally constant uncertainty profile for MCD contrasting with the spatially varying uncertainty of GP and BNN do not support this equivalence in practice for the problems studied.

Regularization techniques (L2, L1, dropout during training) will impact MCD's predictions and uncertainty. However, even when treating these as hyper parameters and informing the loss with the shot noise, the MDC provided variance did not resemble the shot noise.

Although our experiments showed that the MCD method struggles to reproduce the uncertainty, further experiments are needed before the method can be fully rejected. Nevertheless, caution is strongly advised when interpreting its predicted uncertainty.

## 6 Conclusion

We have investigated the Monte Carlo dropout method's ability to approximate the true uncertainty in simple regression tasks, comparing it against Gaussian Processes and Bayesian Neural Networks. While the original paper reported strong performance [1], we find that MDC failed to accurately represent the uncertainty in its posterior predictions for simple example problems. It struggled to capture uncertainty during extrapolation and interpolation, sometimes predicting lower uncertainty than in regions with training data, which was opposite behavior of the BNN and GP benchmarks. Hence, we caution epistemic interpretation of MCD output without empirical validation.

## Author contributions

AD: Formal analysis, investigation, writing - original draft, review and editing, visualisation. AJS: Conceptualisation, methodology, writing - review and editing, supervision. SRS: Conceptualisation, methodology, writing - review and editing, supervision. This paper has taken advantage of NotebookLM[2] to condense sections of a master thesis into the appropriate NLDL format. All text and cited references have been manually checked by the authors. This publication has been partly funded by the SFI NorwAI (Centre for Research-based Innovation, 309834) and the PhysML project (338779). The authors gratefully acknowledge the financial support from the Research Council of Norway and the partners of SFI NorwAI.

---

[2]https://notebooklm.google.com/

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

# A Appendix

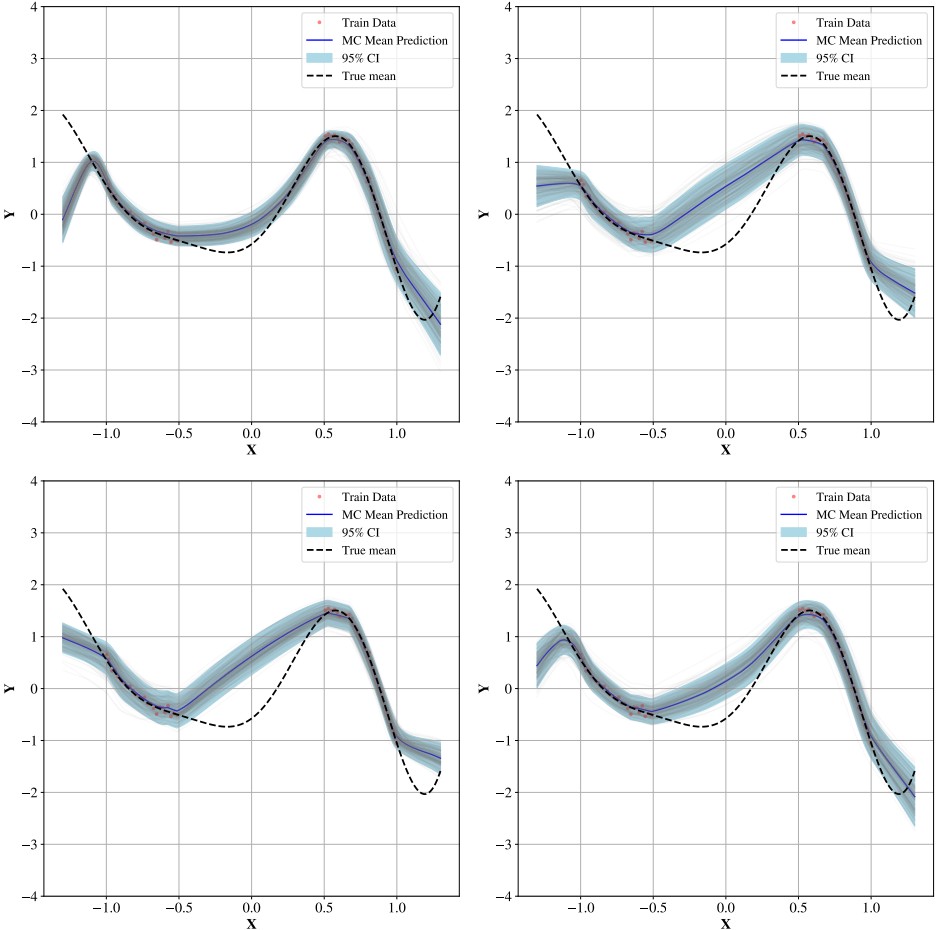

**Figure A.1.** MCD results for different random initialisations of the neural network fitted to 150 data points with noise $\sigma = 0.05$.

