# OpenReview forum: "Unreliable Monte Carlo Dropout Uncertainty Estimation"
_NLDL.org/2026/Conference — NLDL 2026 Poster_

### Official Review · Reviewer_FpF9 · 2025-09-18

**Rating:** 4
**Confidence:** 3
**Final Rating:** 4
**Final Confidence:** 4

**Summary:**

The authors find that Monte Carlo Dropout (MCD) struggles to accurately reflect the underlying true uncertainty associated with BNNs, particularly failing to capture increased uncertainty in extrapolation and interpolation regions observed in Bayesian models.

**Strengths:**

The findings are fraught with consequences for the field of probabilistic machine learning, as a widely-used technique like MCD seems to exhibit a flaw that was not detected before, and so it should probably not be used in future research.

**Weaknesses:**

I believe that the inability of MCD to capture uncertainty correctly is related to the struggle that BNNs generally incur in quantifying uncertainty, especially of epistemic (i.e. reducible) nature in prediction tasks.

To see this, notice that the traditional Bayesian paradigm posits that epistemic uncertainty is all captured by the parameter (prior and/or posterior) distributions. This is a somehow agreeable premise, akin to a second-order approach to uncertainty. When it comes to prediction, though, BNNs use the posterior predictive distribution $p(\tilde{y} \mid \tilde{x}, z_n)$, where $\tilde{x}$ is the new input, $\tilde{y}$ is its related output, and $z_n$ is the training set. As we know, $p(\tilde{y} \mid \tilde{x}, z_n) = \int_\Theta p(\tilde{y} \mid \tilde{x},\theta) p(\theta \mid z_n) \text{d} \theta$, where $\Theta$ is the parameter space, $p(\tilde{y} \mid \tilde{x},\theta)$ is the likelihood, and $p(\theta \mid z_n)$ is the posterior. But it is immediate to see that $\int_\Theta p(\tilde{y} \mid \tilde{x},\theta) p(\theta \mid z_n) d \theta = \mathbb{E}_{\theta \sim p(\theta \mid z_n)} [p(\tilde{y} \mid \tilde{x},\theta)]$, so all the (parameter) epistemic uncertainty encoded in the (parameter) posterior gets "washed away" by taking the expectation. As a result, a single BNN is unable to gauge EU well. I think MCD only exacerbates this problem.

This was already pointed out in https://openreview.net/forum?id=4NHF9AC5ui, and it is a reason why the field of Imprecise Probabilistic Machine Learning is of increasing interest.

If the authors share this opinion, I suggest them to add a sentence pointing this out, and refer the to the works of Hüllermeier, Sale, Rodeman, Caprio, Chau, Muandet, Cozman, de Campos, and others for future research.

**Final Justification:**

The authors answered satisfactorily to my concerns, and the other reviewers seem positive as well.

**Justification:**

The paper exposes a flaw in a widely used technique in Bayesian machine learning, that should be known by the community.

---

> ### Author Rebuttal · Authors · 2025-10-23
>
> We deeply appreciate the reviewer’s thoughtful and philosophically grounded reflections on epistemic uncertainty. The observation that the limitations of MCD may originate from a broader epistemic challenge within Bayesian neural networks (BNNs) themselves adds important conceptual depth to our work. In particular, the reminder that the posterior predictive distribution may attenuate parameter uncertainty through expectation—thereby “washing out” epistemic variability—aligns closely with our empirical observations.
> We concur that MCD, by introducing stochastic subnetworks without true posterior diversity, may in fact exacerbate this limitation rather than resolve it. We will incorporate a dedicated discussion acknowledging this connection between our findings and the theoretical critique of epistemic representation in BNNs.
> We are also grateful for the suggestion to reference ongoing work in Imprecise Probabilistic Machine Learning, including contributions by Hüllermeier, Sale, Rodemann, Caprio, Chau, Muandet, Cozman, and de Campos. By situating our work within this emerging framework, we hope to clarify that the challenge is not restricted to dropout, but reflects a wider need for representations capable of expressing epistemic indeterminacy beyond precise Bayesian expectation.
> We thank the reviewer once again for highlighting these conceptual dimensions, which will help strengthen the framing and impact of the revised manuscript.

---

### Official Review · Reviewer_AGjx · 2025-09-21
**suggest to accept**

**Rating:** 4
**Confidence:** 5

**Summary:**

This manuscript tries to study reliable uncertainty estimation for machine learning models based on traditional Monte Carlo dropout, where you compare to Gaussian Processes and Bayesian Neural Networks, which are popular in nowadays various big data-driven applications

**Strengths:**

1. this draft is easy to understand and follow
2. your claims of MCD are interesting and may have the potential for large-scale data expecially distributed scenarios

**Weaknesses:**

1. your toy problem data is not massive, you're encouraged to significantly enlarge the data size to see the real practical value with more impressive experimental results
2. there are related state-of-the-art you may want to compare: Distributed Clustering of Linear Bandits in Peer to Peer Networks, Fast Distributed Bandits for Online Recommendation Systems

**Justification:**

see above, overall, it's enjoyable to go through this submission, though there are some small jobs need to be done to better polish your merits, in short, it would be glad to recommend this work towards acceptance

---

> ### Author Rebuttal · Authors · 2025-10-23
>
> We sincerely thank the reviewer for their positive evaluation, appreciation of clarity, and recommendation for acceptance. We are pleased that the broader implications of this work were recognized, including its relevance to scalable and applied machine learning.
> We acknowledge the suggestion to extend experiments beyond synthetic data. The 1D setting was chosen as a diagnostic lens to expose fundamental behavior, but we agree that demonstrating these effects in multidimensional or real-world contexts would strengthen practical relevance. In revision, we plan to include additional experiments to illustrate that the observed miscalibration persists even as the data grows more complex.
> We also appreciate the suggestion to connect with distributed and bandit methodologies. While our focus remains on epistemic uncertainty estimation rather than large-scale optimization, we agree that unreliable uncertainty can directly impact decision-making systems, including recommendation and online learning frameworks. We will reference related literature to emphasize these broader implications.
> We are grateful for the reviewer’s recognition that the paper is readable and insightful. We will further refine presentation to ensure clarity while reinforcing the central message: MCD can be useful, but its uncertainty estimates must be interpreted with care, especially in high-stakes or distributed applications.

---

### Official Review · Reviewer_3z4w · 2025-09-27
**Is Monte Carlo Dropout a reliable (well-calibrated) uncertainty estimation method?**

**Rating:** 1
**Confidence:** 4
**Final Rating:** 2
**Final Confidence:** 4

**Summary:**

This paper looks into the extend to which Monte Carlo Dropout (MCD) uncertainty estimation is reliable or well-calibrated in comparison to Gaussian Processes (GPs) and Bayesian Neural Networks (BNNs) on a toy 1D regression dataset. The authors argue that while MCD is widely used due to its efficiency, it does not adequately capture true epistemic and aleatoric uncertainty compared to GPs and BNNs (on this particular regression dataset). MCD produces a nearly constant variance across input regions, failing to reflect the expected increase in uncertainty during interpolation and extrapolation. The introduction frames the study within the need for trustworthy uncertainty estimates in safety-critical machine learning applications.

**Strengths:**

* The motivation for the paper is clear (e.g., why we use uncertainty estimation)
* Connection to standard Bayesian methods (GPs, BNNs) are clearly highlighted.
* The Figures in the results do clearly show a difference in uncertainty estimation between MCD and BNN and GPs.
* The research question/what the paper wants to achieve is clear, e.g., does the theoretical grounding of MCD translate to well-calibrated uncertainty in practice?

**Weaknesses:**

* This work is quite incremental, and the only evidence for the claim of MCD's limitations are results on a toy regression dataset (additionally arguably similar experiments have been done in other work including the work by Y. Gal and Z. Ghahramani). Suggesting that MCD uncertainty estimation is unreliable based on such a dataset is not that convincing.
* It is not entirely clear why there is a discrepancy between the theory (MCD is an approx. to deep GPs) and the empirical results.
* Regarding the evaluation, I expect more rigorous uncertainty estimation evaluation methods/metrics to be used, including proper scoring rules such as expected calibration error and the use of calibration plots (~shows how well a model's predicted uncertainty intervals match the actual frequency of observed outcomes, with perfect calibration lying on the diagonal).
* While related work is cited, the introduction could more clearly identify what exactly has been missing in previous evaluations of MCD and why this study is uniquely positioned to address it.
* Some details in the experimental setup are missing. For example, it is not made clear exactly which GP hyperparameters were optimized when minimizing the negative marginal log likelihood or what hyperparameters were optimized using Optuna.
* Why exactly is the noise variance added to the loss function? This seems like quite a nonstandard approach.
* When comparing MCD and Bayesian NN the same network architecture was not used.

**Final Justification:**

The paper raises an important and clearly stated question about the reliability of Monte Carlo Dropout (MCD) for estimating epistemic uncertainty. The reviewers appreciated the clarity, motivation, and visual evidence, though they noted the lack of quantitative calibration metrics, limited scope beyond synthetic data, and missing reproducibility details.

The authors’ rebuttal addresses at least one of my concerns, and will address the disrepancy between the thoeretical claim about MCD and the emperical results in the paper.

Despite these improvements, the work remains preliminary without the additional experiments and metrics. It presents a valuable cautionary insight but does not yet meet the threshold for acceptance.

**Justification:**

This paper raises an important question about the reliability of MCD for uncertainty estimation and is motivated clearly, with comparisons to other Bayesian methods (GPs, BNNs). However, the contributions is quite incremental, and the evidence is limited to a toy regression dataset, which makes the claim that MCD's uncertainty estimation is unreliable not as convincing. The evaluation lacks rigorous metrics such as calibration plots or proper scoring rules, and some methodological details (e.g., GP hyperparameter optimization, Optuna search space) are missing. The discrepancy between MCD’s theoretical grounding and empirical behavior is acknowledged but not explained, and the BNN comparison is weakened by the use of a different architecture. While the figures highlight differences between methods, the study does not go far enough to support strong conclusions.

---

> ### Author Rebuttal · Authors · 2025-10-23
>
> We thank the reviewer for their critical engagement, recognition of the clarity of our research question, and insightful suggestions. While the study may, at first glance, appear incremental due to its use of a simple 1D dataset, our contribution lies in demonstrating a foundational limitation: that MCD can fail to capture epistemic uncertainty even under ideal conditions. This is not a trivial observation, but a significant warning. If a method fails when correctness should be trivial, its reliability cannot be assumed in more complex scenarios.
> The reviewer rightly points to the discrepancy between theoretical claims that dropout approximates deep Gaussian processes (DGPs) and the empirical miscalibration we observe. In response, in the updated version we will explicitly discuss recent theoretical critiques that question the Bayesian interpretation of dropout. Works such as Le Folgoc et al. (2021), Osband et al. (2016), and Hron et al. (2018) reveal structural limitations—such as improper variational families and risk-focused behavior—that support our empirical findings. We will verify all citations carefully before inclusion.
> Our message is not that dropout should be abandoned, but that its epistemic interpretation must not be assumed without empirical validation.

---

### Official Review · Reviewer_gFgf · 2025-09-29
**Interesting observation, but can be improved**

**Rating:** 2
**Confidence:** 3
**Final Rating:** 2
**Final Confidence:** 4

**Summary:**

This paper investigates the reliability of uncertainty estimates produced by Monte Carlo Dropout (MCD) in the context of Bayesian deep learning. The authors use a simple yet effective 1D regression task where the training data excludes the central region, creating a "data gap." This setup allows them to examine whether different uncertainty estimation methods, including Gaussian Processes (GP), Bayesian Neural Networks (BNNs), and MCD, appropriately reflect increased uncertainty in regions with no data. Their findings show that while GP and BNN models produce wider predictive intervals in such regions, MCD fails to do so, often displaying flat and overconfident uncertainty regardless of input distribution.

Moreover, the paper highlights that the predictive uncertainty of MCD is highly sensitive to the model’s random initialization (i.e., the random seed). Through multiple trials, the authors show that MCD results vary significantly across seeds, and only when aggregating predictions from many different seeds does the uncertainty pattern begin to resemble that of GP or BNN. These findings suggest that MCD, in its common implementation, may be unreliable for capturing epistemic uncertainty, particularly in low-data or out-of-distribution scenarios. The authors conclude with a cautionary message for practitioners who rely on MCD for safety-critical applications.

**Strengths:**

1. The paper addresses a clear and important question: Does Monte Carlo Dropout (MCD) provide reliable uncertainty estimates, especially in regions with little or no data? This is a highly relevant issue in real-world applications such as robotics, healthcare, and safety-critical systems. The clarity and significance of the research goal strongly enhance the motivation for the study.
2. The use of a 1D regression task with a deliberately created data gap is a simple yet powerful approach. This design enables controlled investigation of epistemic uncertainty without the confounding factors present in high-dimensional or real-world datasets. It contributes positively to the paper’s soundness and interpretability.
3. The authors provide well-constructed visual comparisons between MCD, GP, and BNN. These figures clearly demonstrate the discrepancy in uncertainty behavior across models, especially in the data gap region. Such visual clarity aids understanding and effectively supports the authors’ claims.
4. A notable strength is the analysis of how the random seed impacts MCD’s uncertainty estimates. This draws attention to the often-overlooked instability of stochastic methods like MCD, and adds robustness to the conclusion that MCD may be unreliable without additional precautions (e.g., ensembling across seeds).

**Weaknesses:**

1. The paper primarily uses MSE and visual inspection to compare uncertainty estimates across models. While the visualizations are helpful, the absence of standard calibration and sharpness metrics such as Negative Log Likelihood (NLL), Expected Calibration Error (ECE), or coverage versus interval width severely limits the rigor of the evaluation. As a result, the argument that MCD is unreliable remains largely qualitative.
2. All experiments are performed on a synthetic 1D regression task. Although this setting enables controlled comparison, it limits the generality of the findings. The results may not extend to higher-dimensional or real-world datasets, which often exhibit more complex uncertainty structures. Without broader validation, the paper’s conclusions remain narrow in scope.
3. The paper fixes MCD to a specific architecture and dropout rate (e.g., $p = 0.15$) without sufficient ablation on key design choices. Important factors such as the number of Monte Carlo samples, dropout placement (input vs. hidden layers), and the impact of training-time vs. inference-time dropout are not analyzed. This makes it difficult to conclude whether the observed unreliability is due to MCD itself or suboptimal configurations.
4. While some implementation details are provided, the paper lacks critical information necessary for full reproducibility: precise random seed values, network weight initialization strategies, learning rate schedules, and the exact configuration of the Optuna search space. Additionally, no code or data generation scripts / or more information are shared. This limits transparency and hinders reproducibility, especially given that the main findings are empirical.

**Final Justification:**

Given the rebuttal, I'm still leaning slightly toward rejection. However, I wouldn't strongly oppose the decision if the paper were to be accepted.

**Justification:**

The paper presents a clearly motivated and well-designed empirical study that highlights the limitations of Monte Carlo Dropout (MCD) in estimating epistemic uncertainty, particularly in data-sparse or out-of-distribution regions. The authors support their claims with intuitive visualizations and a controlled synthetic setup, and they include an insightful analysis of seed sensitivity. However, the contribution is currently weakened by a lack of quantitative uncertainty metrics, and somewhat narrow experimental scope. Additionally, key details for reproducibility are not fully provided. While the negative result is valuable, the current version does not yet meet the standard for publication without further validation. Therefore, I recommend a (weak) rejection.

---

> ### Author Rebuttal · Authors · 2025-10-23
>
> We sincerely thank the reviewer for the time, effort, and thoughtful evaluation of our work. We are encouraged that the motivation and findings were clearly understood. Our intention is not to argue that Monte Carlo Dropout (MCD) is without practical value, but to highlight that, in its commonly used “out-of-the-box” form, it may fail to represent epistemic uncertainty—especially in regions where no data is available. The goal of this study is cautionary: to urge practitioners to critically examine uncertainty estimates rather than assume that stochastic predictions equate to calibrated epistemic confidence.
> With respect to the use of a 1D synthetic setting, we acknowledge its limited generality, yet this choice was deliberate. By working in a controlled and didactic example where the true uncertainty structure is transparent, we are able to reveal failure modes that would be obscured in high dimensions. If a method cannot reflect uncertainty in such a simple scenario, its behavior becomes even harder to interpret, validate, or correct in more complex settings where ground truth is unknown.
> These two points can be made even clearer in an updated version of the paper.
> We also recognize concerns regarding architectural and hyperparameter variations. We explored alternative dropout rates and placements, but these adjustments did not restore alignment with true epistemic behavior. In practice, since true uncertainty is rarely observable, tuning dropout to “correctness” is infeasible—underscoring the need for caution in interpretation. Regarding reproducibility, we agree fully and will publish all code, scripts, and data generation procedures to ensure transparency.
> We appreciate the constructive critique and will strengthen the manuscript accordingly, including calibration metrics and broader experimental discussion.

---

### Meta-Review · Area_Chair_EUNQ · 2025-11-01

**Recommendation:** Accept (Poster)
**Confidence:** 4

**Metareview:**

This submission investigates the reliability of Monte Carlo Dropout (MCD) as an uncertainty estimation method in deep learning, comparing its behaviour with that of Gaussian Processes and Bayesian Neural Networks in a controlled 1D regression setting. The authors argue that MCD fails to reflect epistemic uncertainty, especially in data-sparse regions, and that its predictive behaviour is sensitive to random initialisation.

The reviewers generally appreciated the paper's clarity, motivation, and visual evidence supporting the claims. The use of a simple synthetic dataset was seen as a strength for isolating the behaviour of uncertainty estimation methods. The authors’ rebuttal was comprehensive and addressed several concerns, including plans to release code and expand the discussion on the theoretical limitations of MCD.

Some concerns were raised about the lack of exploration of key architectural and hyperparameter choices, and about the contributions appearing to be more incremental.

However, I believe the paper would make a good contribution to the conference.

---

### Decision · Program_Chairs · 2025-11-05

**Decision:**

Accept (Poster)

**Comment:**

We recommend a poster presentation given the AC and reviewers recommendations.